# Antibiotic Resistant *Pseudomonas* Spp. Spoilers in Fresh Dairy Products: An Underestimated Risk and the Control Strategies

**DOI:** 10.3390/foods8090372

**Published:** 2019-09-01

**Authors:** Laura Quintieri, Francesca Fanelli, Leonardo Caputo

**Affiliations:** Institute of Sciences of Food Production, National Research Council of Italy, Via G. Amendola 122/O, 70126 Bari, Italy

**Keywords:** antibiotic resistance, dairy products, pseudomonads, spoilage, biofilm, quorum sensing, genomics, control strategies

## Abstract

Microbial multidrug resistance (MDR) is a growing threat to public health mostly because it makes the fight against microorganisms that cause lethal infections ever less effective. Thus, the surveillance on MDR microorganisms has recently been strengthened, taking into account the control of antibiotic abuse as well as the mechanisms underlying the transfer of antibiotic genes (ARGs) among microbiota naturally occurring in the environment. Indeed, ARGs are not only confined to pathogenic bacteria, whose diffusion in the clinical field has aroused serious concerns, but are widespread in saprophytic bacterial communities such as those dominating the food industry. In particular, fresh dairy products can be considered a reservoir of *Pseudomonas* spp. resistome, potentially transmittable to consumers. Milk and fresh dairy cheeses products represent one of a few “hubs” where commensal or opportunistic pseudomonads frequently cohabit together with food microbiota and hazard pathogens even across their manufacturing processes. *Pseudomonas* spp., widely studied for food spoilage effects, are instead underestimated for their possible impact on human health. Recent evidences have highlighted that non-pathogenic pseudomonads strains (*P. fluorescens, P. putida*) are associated with some human diseases, but are still poorly considered in comparison to the pathogen *P. aeruginosa.* In addition, the presence of ARGs, that can be acquired and transmitted by horizontal genetic transfer, further increases their risk and the need to be deeper investigated. Therefore, this review, starting from the general aspects related to the physiological traits of these spoilage microorganisms from fresh dairy products, aims to shed light on the resistome of cheese-related pseudomonads and their genomic background, current methods and advances in the prediction tools for MDR detection based on genomic sequences, possible implications for human health, and the affordable strategies to counteract MDR spread.

## 1. Introduction

Antimicrobial resistance (AR) or multidrug resistance (MDR) is the ability of a microorganism to withstand the action of one or more antimicrobial compounds. Microorganisms that develop MDR are trivially referred to as “superbugs” that are able to tragically worsen the clinical situation of a patient with simple infective episodes and therefore they are a potential threat to human health [1].

In the last few decades, MDR has been elevated into an urgent concern involving global population and a number of relevant sectors, such as clinical, food, and agriculture [2]. Besides the higher rate of mortality in patients with resistant bacteria-related infections, the occurrence of MDR mandates longer hospitalization and therapies with complex and expensive treatment modalities [3]; The Organization for Economic Co-operation and Development (OECD) has predicted 2.4 millions of deaths in Europe, North America, and Australia and a cost up to US$3.5 billion per year in the next 30 years [4]. Despite many efforts and worldwide action to contain the MDR spread [5], the latest data collected by the European Centre for Disease Prevention and Control (ECDC) have unfortunately shown significant increments in the percentages of antibiotic resistant pathogenic bacteria [6]. Among these, *Escherichia coli* and *Pseudomonas aeruginosa* were resistant to third-generation cephalosporin and last resort carbapenem, respectively.

In 2013, the U.S. Center for Disease Control and Prevention (CDC) published a comprehensive report identifying the top 18 antibiotic-resistant threats in the U.S. [7]. *Pseudomonas aeruginosa* was included in the list of serious threat: More than 6700 MDR *Pseudomonas* spp. infections per year have been reported and 440 death per year were estimated to be caused by these species.

This scenario has been foreseen to get worse if novel classes of antibiotics do not reach the market; regretfully, most pharmaceutical companies have ceased or have severely limited investments in the development of new antibiotics due to a high failure rate of past clinical trials, genus and strain-specificity, or adverse effects of molecules [8]. During last years, only 7 novel antibiotics have received marketing authorizations, whilst other molecules in late-stage development are in Phase 2 or Phase 3 of clinical trials [8]; these latter also include mixtures of new and existing classes of antibiotics in order to provide a multiple mechanism and increase their success rate, above all, against MDR bacteria.

In addition to the intrinsic resistance, bacteria can acquire resistance by de novo mutation or via the acquisition of resistance genes (ARGs) from other organisms [9]. In the environment, ARGs, carried by bacterial contaminants, can multiply in their hosts, be shared with other competent bacterial populations (such as human pathogens), and be subjected to further genetic changes [10]. Recent studies suggest that ARGs are acquired through horizontal gene transfer (HGT).

While the vertical gene transfer is the transmission from parent to offspring, the HGT is not inheritance-characterized and allows the exchange of genetic material among more or less evolutionary distant organisms. The HGT may occur via three main mechanisms: Transformation, transduction, or conjugation. The process of gene transfer by transformation does not require a living donor cell but involves the uptake of a short fragment of naked persistent DNA from the environment by naturally competent bacteria. By transduction, DNA is transferred from one bacterium to another via bacteriophages. Conjugation is a sexual mode of genetic exchange and is the only one which requires cell-to-cell contact by conjugative pilus, which provides a physical link for DNA to move between donor and recipient pili (reviewed by [11]). HGT is recognized as the major evolutionary force which has contributed and still drives the shaping and remodeling of bacterial genomes throughout evolution. 

Mobile genetic elements which carry ARGs are generally included in genetic islands, located in plasmid or integrated in the chromosome through the action of integrons and transposons, associated with genes conferring resistance to metals, dye, and biocides, or enzymes providing new metabolic pathways which may contribute to increase the fitness of the microorganism or conferring advantages in the exploitation of ecological niches [12]. 

These genetic clusters show remarkable capability of spreading among several commensals living in the same niche [13]. Well known examples of this horizontal gene transfer (HGT) are found between actinobacteria and proteobacteria also in “carry-back” mechanisms [14]. Together with the misuse of antibiotics, biotic and abiotic factors (such as physicochemical conditions, environmental contaminants, induction of stress responses, bacterial adaptation, and phenotypic heterogeneity) have the potential to enhance the effect of selective pressures and promote bacterial evolution towards antibiotic resistance. 

As further on discussed, although MDR in *Pseudomonas* spp. occurring in diary has not yet emerged as a public health threat, several ARGs have been identified in these species and some of the isolated strains have been recognized as MDR. The transfer of genetic material and ARGs through HGT among bacteria which inhabited the same ecological niche may lead to the spread of AR determinants to pathogenic agents, worsening their eradication from the processing environment, foods, and consumers (Figure 1).

Typically, MDR bacteria are associated with nosocomial infections. Among these bacteria, *P. aeruginosa* is the species most frequently associated with diseases in hospitalized patients, where it acts as an opportunistic pathogen causing infections in almost any organ or tissue, especially in patients with a weakened immune system [15]. Moreover, recent evidences highlighted that non-pathogenic pseudomonads strains can cause bacteremia in humans [16]; they have also displayed several MDR, gained from HGT or inheritable mutations: This might pose a severe threat to human health due to their high versatility and adaptation [17]. Most members of this group are psychrotrophic, generally occurring in water and soil or associated with plants [18,19]; however, they also naturally contaminate both fresh foods, such as dairy products, and their processing environment, and they are mostly feared for their spoilage capabilities [20,21,22,23]. In minimally processed fresh foods these bacteria are not included among the hygiene and safety regulatory microbial targets [24]; in addition, in the absence of adequate disinfection practices, they are even tolerated at high loads in fresh processed dairy and meat products [20,25,26]. Indeed, food-associated pseudomonads are commonly considered commensal bacteria, but they might transfer AR genes to human pathogenic bacteria during food processing or after ingestion, raising possible risks for human health [27].

Furthermore, it has been demonstrated that the predominance and persistence of pseudomonads in foods and on surfaces of food processing plants is related to the ability of these microorganisms to form biofilm, which enhances their resistance to adverse conditions including several antimicrobial treatments [21,28,29]. Psychrotrophic biofilm-forming strains, endowed with the ability to withstand temperature changes, to counteract reactive oxygen species and nutritional starvation, might have an advantage in comparison with other thermoduric microorganisms [30]; not surprisingly, pseudomonads have become the dominant population in several foods (such as milk and fresh cheeses; [20]) and in food production environments [28]. Therefore, the finding of psychrotrophic MDR pseudomonads within the specific group of spoilage organisms, generally ingested alive through refrigerated fresh dairy products, poses a number of issues which are totally overlooked. Their possible spread routes and implications of antibiotic resistance of these foodborne bacteria are schematically shown in Figure 1.

The present review addresses the main aspects related to the persistence and spread of MDR *Pseudomonas* on fresh dairy products, their genomic background, the prediction tools for MDR detection based on genomic sequences, and possible implications and strategies for their control.

## 2. *Pseudomonas* spp. Genus and Species Occurrence in Dairy Products

At the present time (July 2019) the genus *Pseudomonas* (family *Pseudomonadaceae,* class *Gammaproteobacteria*) comprises 272 validly identified species (according to the List of Prokaryotic Names with Standing in Nomenclature http://www.bacterio.net/) among which 114 were represented by at least 1 type strain (WFCC Global Catalogue of Microorganisms; http://gcm.wfcc.info/overview/). It is considered the most heterogeneous group of Gram-negative bacteria including saprophytic aerobic rods and motile, catalase-positive, and non-spore-forming bacteria [31]. Moreover, some species can grow anaerobically, using nitrate as an electron acceptor. Depending on species, pseudomonads encompass psychrotolerant or psychrotrophic species (growing below 15 °C) and thermophilic species like *P. aeruginosa* (growing up to 42 °C) [19,32]. 

Due to their simple nutritional requirements and their high metabolic versatility, these bacteria are ubiquitous, having been isolated from a variety of sources (soil, fresh water, humans, plant and animal surfaces, cosmetics, medical products and instruments, foods). Some of these bacterial groups act as opportunistic pathogens causing several infectious diseases in animals and humans [33,34]. They also play a role as phytopathogens [35] and as specific spoilage organisms [22,36,37]. In particular, pseudomonads are involved in off-flavor release, due to the production of volatile compounds and amino acid metabolites; in addition, they can produce thermotolerant proteolytic and lipolytic enzymes that heavily reduce quality and shelf-life of proteinaceous foods (i.e., milk, fresh dairy, and meat products; [22,38]. Spoiled foods, especially dairy products, often appear discolored due to the biosynthesis of pigments by some pseudomonad species [39,40,41,42]. *Pseudomonas* spp. can be also endowed with pectinolytic activity causing browning of minimally processed vegetables [43,44]. Unlike the other thermotolerant spoilage microorganisms, which are not able to adapt and grow at low temperature and under oxidative stresses, psychrotrophic pseudomonads can quickly colonize chilled foods causing spoilage and biofilm formation [21,45,46]. This latter mechanism improves their adaptability and spreading, making their eradication very challenging and enhancing their drug resistance [47]. Because of these characteristics, the occurrence of some psychrotrophic pseudomonads (i.e., *P. fluorescens*, *P. fragi*, *P. lundensis*, *P. gessardii*, and *P. taetrolens*) in refrigerated fresh dairy products, as well as in potable water used as governing liquid, has been gaining increasing interest [20,48,49].

The economic burden due to *Pseudomonas* spp. spoilers is roughly one-third of the edible parts of produced food that get lost or globally wasted. Depending on the product types, the dairy sector accounts losses for about 25–30%; in the industrialized countries 50% of these losses are related to the consumption stage, which, in particular in the United States, account for USD $13 billion. Since 70% of the spoiler psychrotrophic isolated from cold stored dairy products (including raw milk) belongs to *Pseudomonas* species, large part of food losses at the consumption stage can be attributed to these bacteria. Their elevated spread in processing environments, from which they can be easily transferred to food at any stage of production, was correlated to the ability to tolerate a wide variety of conditions, including the exposure to conventional sanitizers [50].

The taxonomy of *Pseudomonas* genus was deeply revised in the last two decades. Indeed, firstly, phenotypic characterization methodologies [51] and RNA/DNA hybridization [51,52], then, a variety of molecular approaches (i.e., Repetitive Extragenic Palindromic PCR, Restriction Fragment Length Polymorphism, Denaturing Gradient Gel Electrophoresis, Pulsed Field Gel Electrophoresis, and 16S rDNA locus sequencing [45,53,54]) has led to many changes and reclassification to the taxonomy of pseudomonads. Finally, the genus *Pseudomonas* (*sensu stricto*) was reorganized in 7 different subclusters (*P. syringae*, *P. chlororaphis*, *P. fluorescens*, *P. putida*, *P. stutzeri*, and *P. aeruginos*a) basically consistent with the phylogenetic classification of Palleroni et al. [55], among which *P. aeruginosa* group was the most distinctive [56,57,58]. However, due to its high conservative degree, the 16S rDNA gene sequencing did not allow an optimal resolution of certain closely related *Pseudomonas* species [36,56,59]. Therefore, multilocus sequence analysis (MLSA) or multilocus sequence typing (MLST) of housekeeping genes (16S rRNA gene, *gyrB*, *rpoB*, *rpoD*) has been increasingly used for discriminating pseudomonads species within phylogenetic studies [32,60]. MLST has proved to be a powerful tool for characterizing bacteria based on allelic variations of housekeeping genes. Interestingly, some of these genes (i.e., caseinolytic metallo-protease gene *aprX* [61]) and targeted MLST phylogenetic schemes (based on an housekeeping sequences uploaded to the *Pseudomonas* MLST databases https://pubmlst.org/ and http://microbiologia.uib.es/bioinformatica/index.html) have been very effective at discriminating and tracing psychrotrophic pseudomonads involved in proteolysis of spoilage cheeses [62,63] and fresh cheese discoloration [64]; in particular, the blue discoloration of different fresh cheeses (such as high moisture mozzarella cheese, Latin style low acid fresh cheese, and ricotta cheese), recognized as a worldwide problem, was attributed to a “blue branch” of the *P. fluorescens* phylogenetic tree [39,40,41,42,43,44,45,46,47,48,49,50,51,52,53,54,55,56,57,58,59,60,61,62,63,64,65].

Other techniques such as the time consuming PFGE, as well as other genotyping methodologies (random amplification of polymorphic DNA analyses (RAPD), enterobacterial repetitive intergenic consensus (ERIC), and amplified fragment length polymorphism (AFLP)) have efficiently allowed the exploration of genetic variability among *Pseudomonas* populations from different sources, although without taking account of the viable but non-culturable (VBNC) bacteria or of the genes directly associated with the spoilage traits of food-related pseudomonads [63,64,65,66]. 

Only ten years ago, based on MLSA from four housekeeping genes (16S rRNA, *gyrB*, *rpoB*, and *rpoD*) of the type strains of 107 of *Pseudomonas* spp. [32], grouped pseudomonads into two main separate lineages. That assigned to *P. aeruginosa* appeared homogeneous even if it contained three main groups (*P. aeruginosa*, *P. stutzeri*, and *P. oleovarians*). The *P. fluorescens* lineage harbored six groups (*P. fluorescens*, *P. syringae*, *P. lutea*, P*. putida*, *P. angiulliseptica*, and *P. straminea)*. The *P. fluorescens* species group was the most variable, including nine subgroups, among which the *P. fluorescens* subgroup that comprised a further 20 distinct species names, is predominant in dairy products. [67] described the *P. fluorescens* group based on MLSA and whole-genome sequence-based analyses of 93 sequenced strains, dividing the complex in eight phylogenomic groups, consistent with the digital DDH analysis. 

The psychrotrophic species belonging to *P. fluorescens* lineage were also designated as Specific Spoilage Organisms (SSO) because of their strong food spoilage potential due to the production of specific enzymes, pigments, and off-flavors [56,68,69,70,71]. 

Recently, the combined approach of both “taxonomic oriented” (or metagenetic) and “oriented function” (or metagenomic) techniques was suggested to fingerprint microbial communities directly related to food quality, spoilage, or safety and without the need of cultivation on synthetic microbiological substrates [72,73]. Recent applied examples in cheese have led to the determination of the composition of microbial consortia and their functional role in flavor development [74,75] by unequivocally identifying bacterial species and metabolic pathways responsible for cheese discoloration [76]. These first results could be also used for profiling spoilage psychrotrophic pseudomonad populations associated with refrigerated fresh dairy products by generating gene libraries aimed at understanding several metabolic functions.

## 3. Antibiotic Resistance in *Pseudomonas* spp. Spoiler: Mechanisms and Influencing Factors in Dairy Sector

Neglecting the human pathogen *P. aeruginosa*, in the last decades, foodborne *Pseudomonas* spp., especially isolated from cold stored products, have been found to be resistant to different classes of antibiotics [25,77]. Table 1 and Table 2 show a list of resistant *Pseudomonas* spp. strains isolated from dairy products and related ARGs, respectively. However, in order to emphasize the severity of the problem, AGRs identified from *Pseudomonas* spp. naturally occurring in other foods or in the environment are included in Table 2 and briefly discussed throughout the paper. 

Several species of dairy-borne *Pseudomonas* spp. are resistant to different β-lactams, belonging to the four structural classes (penicillins, cephalosporins, carbapenems, and monobactams; [78] and in combination or not with β-lactamase inhibitors (Table 1). As widely described, these antibiotics, targeting the transpeptidase enzymes involved in cross-linking the peptidoglycan cell, can be inactivated by the synthesis of metallo-β-lactamases as well as by cell wall modifications and changes to the outer membrane architecture to create an impermeable barrier. 

Metallo-β-lactamases are predominantly plasmid encoded as part of mobile genetic cassettes, which facilitates their transmission throughout microbial populations [79]. In past, they were found to mediate resistance in nosocomial infections induced by *P. aeruginosa* [80]. However, acquired class B enzymes of the VIM and IMP Type metallo-β-lactamase were also found in MDR *P. putida* and carbapenem-resistant *P. fluorescens* isolated from hospitalized patients [81,82], whilst the production of an intrinsic metallo-β-lactamase POM-1 was revealed in carbapenem resistant *P. otitidis,* isolated from fresh food [83]. 

As above mentioned, β-lactamase enzymes (acquired and intrinsic) represent only part of the cellular mechanisms that these bacteria developed to counteract antibiotic activity (Table 2). In foodborne *P. putida* strains, carbapenem resistance has been attributed to the overexpression of the TtgABC efflux system as well as to the loss of porins [83].

Bacterial efflux pumps, categorized as primary (driven by ATP hydrolysis) or secondary (driven by proton motive force) and grouped into five major superfamilies [88], can capture and extrude many structurally diverse antibiotics, in addition to non-antibiotic compounds, such as dyes, biocides, or metal ions. Resistance can involve all these major families; in Gram-negative bacteria, the MDR phenotype is largely conferred by resistance-nodulation-cell division superfamily (RND) efflux systems contributing to the intrinsic resistance of *P. aeruginosa* (MexAB-OprM and AcrAB-TolC; [89]), *P. fluorescens* (EmhABC; [90]), and *P. putida* (TtgABC; [91]).

As concerns membrane, fluidity, and composition (fatty acids and proteins components, including RND efflux pumps) can be affected by physical and chemical stresses [92]. Changes in the resistance of cells grown at different temperatures or subjected to various environmental stresses have been reported [93,94]. Recently, EmhABC, an RND-type efflux pump involved in the extrusion of hydrophobic antibiotics (i.e., aminoglycosides) in a *P. fluorescens* cLP6a, changed substrate efflux depending growth temperature [90]. In particular, increased substrate efflux was measured in cLP6a cells grown at 10 °C and decreased efflux was observed at 35 °C compared with cells grown at the optimum temperature of 28 °C. These results suggested that antibiotic efflux systems could be involved in mechanism of adaptation to cold temperature, occurring even without the selective pressure of antibiotics.

The role of temperature in the induction or repression of resistance mechanisms were also suggested by some results obtained in our laboratory. Indeed, a comparative proteomic analysis of *P. fluorescens* ITEM 17298, responsible for anomalous blue discoloration of mozzarella cheese [41], revealed that the macrolide export protein MacA was induced at lower growth temperature, whilst putative efflux pump TtgBC, the MDR protein MdtA precursor, and the bifunctional polymyxin resistance protein ArnA were up-regulated at higher temperatures [21]. The occurrence of antibiotic resistant determinants in this target spoiler suggested that we evaluate the antibiotic resistance of other *Pseudomonas* spp., isolated from high moisture mozzarella cheese [20,41]. In this regard, for the first time, the results of the antibiotic susceptibility tests of several *Pseudomonas* strains are reported in Table 3. Statistical analyses showed that there was a significant difference (*p* < 0.05) in antibiotic susceptibility among the strains. All selected strains were found resistant to β-lactam class penicillins and cephalosporins. In addition, *P. fluorescens* ITEM 17298 was also resistant to the aminoglycoside gentamicin. Antibiotic resistance of the latter is supported by the occurrence of genetic determinants (*mexA, ampC, oprD, mdtL, emrB*) identified by genomic analysis of this strain [95]. 

Other than HGT, bacteria can acquire resistance to antibiotics through mutations of genes regulating specific functions, such as DNA metabolism, translation and cell wall biogenesis; in particular, point mutations in *gyrA* and *gyrB* (DNA gyrase subunits), and *parC* and *parE* (topoisomerase IV subunits) associated with resistance to fluoroquinolone antibiotics have been reported in *P. putida* strains [126]. Since, in the absence of the antibiotics, these mutations can become detrimental for the bacteria, additional mutations, known as “compensatory mutations” can occur; these latter allow antibiotic resistance to become stabilized in the population without further fitness costs. As concerns pseudomonads, compensatory mutations were reported for rifampicin-resistant pathogenic [137], and not pathogenic species isolated from creek sediments [138]. 

In dairy plants, *Pseudomonas* spp. contamination is attributed to contaminated water or pipe surfaces [139]. On piping and fitting surfaces, as well as manufacturing and processing tools, pseudomonads grow as biofilm, a physiological state that confers them an increasing persistence and resistance to conventional sanitizers and antibiotics. *Pseudomonas* biofilm occurs through a cell-to-cell communication (*quorum sensing*, QS) regulating its sequential phases and governing metabolic activities of this ecological niche. In biofilm state, antibiotic resistance can occur through different mechanisms which include (a) the failure of antimicrobials to penetrate throughout the biofilm matrix, (b) a wide range of non-specific protective adaptations (such as enhanced efflux) associated with the biofilm phenotype, and (c) the formation of a persisting cell sub-population [140]. The increased resistance of biofilm phenotype was highlighted by Molina et al. [126] reporting the antibiotic concentrations necessary to eradicate biofilm (rifampicin, fluoroantimonic, amikacin, ceftriaxone, and norfloxacin) are from 3- to 40-fold higher than those required to inhibit planktonic cells. Biofilm-specific antibiotic resistance and tolerance is multifactorial, and related mechanisms vary depending on the antibiotic, the bacterial strain and species, the age and developmental stage of the biofilm, and the biofilm growth conditions. Among factors affecting the biofilm phenotype, a positive correlation between low temperatures and biofilm production was demonstrated in foodborne *P. fluorescens* [21,141]. 

In order to highlight the role of temperature in bacteria adaptation and persistence, we reported the effects of two growth temperatures on some phenotypic traits of *Pseudomonas* spp. isolated from fresh cheese (Figure 2). Biofilm biomass, cellulose production and motility increased at low temperature suggesting that adaptive mechanisms, underlying physiological changes, need to be further investigated. In addition to several spoilage phenotypic traits, the pigment biosynthesis is also induced by low temperature (Figure 2; [21,141]). Pigment synthesis is putatively orchestrated to counteract the increased oxidative stress that the spoilage pseudomonads undergo at low temperatures [21,142]. In this regard, an oxidative stress sensing and response *ospR* (oxidative stress response and pigment production regulator) gene was found in *P. areuginosa*. This gene, under the control of the quorum sensing (QS) system, also affects *P. aeruginosa β*-lactam resistance, strengthening the hypothesis that the exposure to certain levels of oxidative stress may switch on defensive pathways in *P. aeruginosa,* thus making it more resistant to killing by immune cells [143]. Likewise, in a blue cheese pigmenting *P. fluorescens,* oxidative stress-regulators (OxyR) and metabolites (polyamines) increased their amounts under low temperatures [21]. The latter were previously found to be correlated with the increase of antibiotic resistance in other microorganisms in the absence of selective pressure [144,145]. These cellular mechanisms suggest that the spoilage microbiota is able to modulate its own metabolic work in order to enhance its adaptation and competitive advantage against other food-associated bacteria [109,146].

## 4. Sequencing-Based Tools and Database for AR Prediction 

AMR is traditionally studied by standardized phenotypic assays based on antimicrobial susceptibility testing methods. Faster and less variable molecular methods have also been used for surveillance and to support clinical therapy. Recently, whole genome sequencing (WGS) based methods have become a broad branch of AMR analysis, taking the advantage of the ever-lower cost of sequencing, the increasingly availability of genomic data and associated metadata, and the development of dedicated algorithms which have been ensuring a tremendous insight into unknown reservoirs and novel AR genes [148].

The identification of AMR genes from sequencing data has the advantage of not relying on bacterial growth and not passing through experimental work such as the selection of suitable marker genes, the development of standardized protocols, and the interpretation of results. With the availability of a plethora of bacterial genomes and the machine learning approaches, several repository databases have developed and integrated tools for ARM prediction and annotation. As an example, PATRIC (http://patricbrc.org/) has collected, to date, a total of 202,630 bacterial genomes adding, where available, their ARM phenotype as well as other metadata including minimum inhibitory concentrations. The PATRIC database also provides tools to predict and identify genomic regions related to AMR [149] based on computational analysis. These approaches rely on machine learning algorithms developed by selecting the oligonucleotide *k*-mers relevant to AR [150,151], which are then used as a phenotype classifier towards unknown genomes. PATRIC also integrates knowledge of known AR genes from other databases, such as the Comprehensive Antibiotic Resistance Database (CARD, https://card.mcmaster.ca/), the Antibiotic Resistance Genes Database (ARDB, https://ardb.cbcb.umd.edu/), and the National Database of Antibiotic Resistant Organisms (NDARO, https://www.ncbi.nlm.nih.gov/pathogens/antimicrobial-resistance/).

The Comprehensive Antibiotic Resistance Database [152] comprises (to date) 4071 ontology terms, 2553 reference sequences, 1216 SNPs, 2605 AMR detection models and allows the resistome prediction of 79 pathogens, 4264 chromosomes, 4780 plasmids, 66,402 WGS assemblies, and 146,945 alleles. It includes a collection of resistance determinants and associated antibiotics and tools for analysis of molecular sequences. Analysis implements both the BLAST tool and Resistance Gene Identifier (RGI) software, which predicts resistomes based on homology and SNP models. CARD is an ontology-driven database based on four central ontologies: The ARO (Antibiotic Resistance Ontology), MO (Model Ontology), RO (Relationship Ontology), and a subset of the NCBI Taxonomy Ontology. ARO is the core of CARD and it is a novel controlled vocabulary, which categorizes the information describing antibiotics and their molecular targets, antibiotic resistance genes and mutations, proteins, mechanisms, and associated phenotypes. ARO is integrated in RGI in which terms are associated with bioinformatic models and sequence data in order to predict antibiotic resistance at the genome level. The further strength of CARD is its actively curation which is necessary to keep track of the continuous identification of novel AR due to the antimicrobial pressure favoring new AMR mutations, the transfer among pathogens, and the emergence of novel AMR genes from environmental sources and from the protoresistome.

ARG-ANNOT was the first database, developed in 2014 by Gupta et al. [153], to include detection of point mutations in chromosomal target genes known to be associated with antimicrobial resistance and was built with 1689 antibiotic resistance genes. ARG-ANNOT uses a local BLAST program in Bio-Edit software that allows sequence analysis without a Web interface.

The NDARO database is part of the U.S. National Center for Biotechnology Information and is a collaborative, cross-agency, centralized repository of AR genes and pathogenic organisms. In this case, the software used for AMR prediction is the AMR finder, integrated in a dedicated pipeline for pathogen detection. It identifies resistance genes using as input proteins (applying both BLASTP and HMMER to search protein sequences for AMR genes along with a hierarchical tree of gene families to classify and name novel sequences) or nucleotide sequences (by BLASTX translated searches and the hierarchical tree of gene families). The NCBI’s Bacterial Antimicrobial Resistance Reference Gene Database contains 4000 curated AMR protein sequences. It also allows the submission of novel AMR, pathogen-associated sequences, genome submission and antimicrobial susceptibility testing (AST) data. 

Differences in these software performances are related to the extensiveness of the database they rely on, the length of sequence and the percentage of similarity used as a cut-off, thus affecting specificity and sensitivity of the prediction.

Other than these general AMR databases, there are others that are focused on a specific genus/pathogen or on antibiotic resistance enzymes.

The *Pseudomonas* database [154], since 2000, has contributed to the improvement of the knowledge about the *P. aeruginosa* PAO1 genome with more than 2500 updates by a community-based approach. The database has also integrated tools for comparative genomic analysis and to date comprises a total of 3348 *Pseudomonas* genomic sequences, with 106 of *P. fluorescens* strains, 66 of *P. putida* and other *Pseudomonas* spoilers (3 of *P. fragi*, 1 *P. gessardii*, 1 *P. lundensis* etc.). In the database, the AMR gene predictions results, derived from CARD interrogation, were added for all the strains included in the repository.

Examples of specific AR enzymes databases are the *β*-Lactamase (Database BLAD; www.blad.co.in; [155]) which includes resistance patterns of all class of beta-lactamases as well as the crystal structures of protein data bank (PDK), or the weekly updated beta-lactamase database BLDB (http://www.bldb.eu/; [156]) describing sequence information, biochemical, and structural information on all the currently known *β*-lactamases. BLDB categorizes more than 4326 beta-lactamases based on their class (Class A, Sub-class B1, Sub-class B2, Sub-class B3, Class C, Class D), also offering kinetic values. BLDB also integrates addition tools to analyze *β*-lactamases linking them to the related web resources (NCBI, PDB, etc.).

In AST, the threshold-based assessment scheme of the degree of drug effectiveness is defined by the new ISO 20776-1 standard as “susceptible”, “intermediate”, or “resistant”, depending on the MIC value. The interpretation of phenotypic data is critical, since erroneous categorization of true-susceptible isolates as resistant could lead to unnecessary restriction of therapeutic options. This is particularly relevant for the intermediate category, which makes published data on resistance difficult to meaningfully compare; furthermore, protocols and data interpretation are not standardized for all genera.

Traditional antimicrobial susceptibility testing can underestimate the risk of resistance since they are performed under standardized condition that could not allow the expression of AR genes although present. On the other hand, this is also the risk associated with *in silico* prediction, which can generate an overestimation of the risk due to the presence of genetic elements recognized as involved in AR but not necessary implying that the gene is actively transcribed or expressed, or leads to a true resistance.

Despite traditional, molecular- and WGS-based methods for the detection of foodborne spoiler and pathogenic bacteria [157], AR gene detection methods have recently been developed for the application in clinical specimens by using metagenomic sequencing with customized pipelines [158]. In addition to issues related to technical application, such as experimental costs, sampling size and the implementation of low input protocol, major challenges are due to the presence of poorly distinguished allelic variants. Recently, the application of whole metagenome sequencing (WMS) for screening AMR genes in food and clinical samples has proved very attractive, although the high rate of false negatives and the lack of a detection threshold strongly limits its use [159]. Nevertheless, some studies [160,161] carried out with this approach utilize the strategy to clone the total community of genomic DNA extracted from food samples into fosmid libraries which are then screened onto selective media to identify AMR clones. Shotgun sequencing and annotation of resistant vector allows the detection of AMR genes occurring at low frequency in food matrix. Following this strategy Florez et al. [161] highlighted a risk of HTG associated with tetracycline resistance gene types located in plasmids of lactic acid bacteria occurring during traditional Spanish cheese ripening. Likewise, Devirgiliis et al. [162] put in evidence ampicillin- and kanamycin-resistant clones from *Streptococcus salivarius* subsp. *thermophilus*, and *Lactobacillus helveticus* genomes achieved through the development of a metagenomic library and hosted several open reading frames flanked by highly mobile regions. To date, no study using the WMS approach has been performed to demonstrate lateral AMR gene transfer among food microbiota including commensal or opportunistic bacteria like psychrotrophic food spoilers. However, Berman and Riley [161], analyzing metagenomic plasmid libraries from retail spinach, identified 5 novel AMR genes (like dihydrofolate reductase for sulfonamide class) associated with food saprophytes, including *P. fluorescens*, suggesting they could be considered a reservoir for drug-resistance genes and the transfer to human pathogens. Moreover, Meier et al. [103] identified, by using ARDB, several ARG (Table 3) in the industrial *P. fluorescens* strain ATCC 13525 isolated from pre-filter tanks in England and used as a degrading microorganism in wastewater treatment.

A metagenomic approach was also used by Hendriksen et al. [163] within the Global Sewage Surveillance project to monitor AR in urban sewage by applying ResFinder to whole community sequencing reads. ResFinder software is based on a BLAST alignment of the input sequence file and the specific allele set; it combines ResFinder.pl identifying acquired genes, and PointFinder.py identifying chromosomal mutations. These two programs are based on two different curated databases [164]. It also accepts analysis performed on raw reads.

Although we are quite far from the development of rapid sequencing-based AR detection methods to be applied to food matrices, *in silico* prediction has been proven highly concordant with phenotypic antimicrobial susceptibility [164,165]. Machine learning techniques are also becoming useful to predict novel biomarkers based on the presence of some genotypic elements or mutation predictive of a given phenotype thus leading to the development of diagnostic tools. 

The success of these tools relies on the largeness of available data, their accuracy, and the balance between resistance profiles used. Thus, the publicity of genomic and related metadata is the basis of the future desirable prediction tools.

## 5. Strategies to Control the Spread of Antibiotic Resistant *Pseudomonas* spp. in Dairy Sector 

Psychrotrophic *Pseudomonas* sp. species harbor a pool of resistance genes that can be transmitted to other bacteria, animals, or humans [26]. The threat of a mobile resistome, causing increasing hospitalization and mortality rates of patients, is turning into reality and needs urgent solutions. Food products, where pseudomonads grow and induce spoilage, can be a vehicle accelerating this phenomenon. In response to the conventional methods of food preservation such as cold storage, contaminating *Pseudomonas* spp., (e.g., *P. fluorescens*) have increased their biofilm formation, which determines a higher bacterial tolerance to antimicrobials and represents a constant source of contamination in the environment. Currently, no solutions have been provided yet [166,167] to efficiently counteract *Pseudomonas* spp. spread and persistence in the food sector. The challenge is doubled, as there must also be consideration that control strategies should prevent *Pseudomonas* spread without affecting product quality. 

Recently, among the most promising strategies are natural compounds, including plant extracts, that can be easily used as additives in dairy food preservation. However, most compounds have been tested in vitro assays, mainly aimed at evaluating antimicrobial activity against both *Pseudomonas* spoilers and pathogens [168,169,170,171]. Based on the complexity of the food matrix, as well as intrinsic and extrinsic factors interfering with the activity of these compounds, the results obtained in vitro could not be confirmed in foods. To the best of our knowledge, very few studies have reported the activity of vegetable derived compounds *in situ* trials. Among promising preservatives, phenols from olive oil by-products, herbs, or spices, effectively retarded the growth of *P. fluorescens* and *Enterobacteriaceae* in fresh cheeses [172,173,174] resulting in the prolonging of shelf life and the improvement of sensorial acceptability thresholds. However, no data have been reported on the absence of detectable resistances towards these compounds or on the effect on the expression of ARGs.

In the last few decades, antimicrobial peptides (AMPs) have been suggested as adjuvant or alternatives to antibiotics to counter antibiotic resistance. To date, three peptides of microbial origin (colistin, gramicidin, and daptomycin), have been approved by the Food and Drug Administration (FDA) and used as antibiotics to treat bacterial infections in a clinical setting; advantages in the use of AMPs consist of (a) the broad spectrum of displayed antibacterial, antiviral, anti-parasitic, and anticancer activities, (b) a lower acquired resistance compared to antibiotics, (c) the reduction of induced mutagenesis triggered by bacterial stress pathways, and (d) the synergistic interactions with other antimicrobials [175]. 

In dairy sectors, AMPs have been effectively used to counteract pseudomonads growth and spoilage. In particular, [176,177] investigated the antimicrobial activity of bovine lactoferrin-derived peptides (BLFPs) against *Pseudomonas* spp. strains in order to discover new applications in the dairy sector. The antimicrobial potential of these peptides against food spoilers was suggested by the well-studied properties of bovine lactoferrin protein (BLF), already used in food supplements addressed to human consumption [178], as well as by the presence of stretches with high antimicrobial activity against human pathogens [179]. BLFPs exhibited in vitro a high antimicrobial activity against *P. fluorescens, P. putida*, *P. fragi,* (bactericidal activity of the identified peptide, Lactoferricin B, at 20–80 µM). The antimicrobial activity was also confirmed for the purified Lactoferricin B and its related pepsin-digested BLF hydrolysates (HLF) in cold stored high moisture mozzarella cheese; these molecules reduced the growth of both the naturally contaminating *Pseudomonas* spp. [176,177] and the inoculated *Pseudomonas* target strain in the first days of cold storage [180], also counteracting the appearance of anomalous discoloration [41]. By contrast, no antimicrobial activity was registered against other bacteria, such as the starter lactic acid bacteria (*Lactobacillus delbrueckii* subsp. *lactis* and *Lactococcus lactis*), suggesting the potential usage of these peptides during the manufacturing process of dairy products (*Quintieri, personal communication)*. In light of these results, an antimicrobial LfcinB functionalized coating was developed using cold plasma-based technologies in order to be applied in the packaging of dairy products [181]. 

In an effort to reveal the mechanism of action of HLF against the blue cheese pigmenting *Pseudomonas* strain, the effects on the synthesis of MDR-related proteins were also shown in a comparative proteomic study [21]. Results highlighted that the multidrug transporter MdtA and the macrolide export protein MacA, whose expressions were increased under cold temperature, were inhibited by peptide treatment. By contrast, the transcriptional regulator PhoP, that in *P. aeruginosa* regulates resistance to cationic antimicrobial peptides, polymyxin B, and aminoglycosides [182] increased under treatment; consequently, the bifunctional polymyxin resistance protein ArnA was detected at high levels. 

This result is not unexpected, as mechanisms of resistance to AMPs are widespread in bacteria, included Gram-negative bacteria. Proteolytic degradation, changes in the physical–chemical properties of surface molecules or the cytoplasmic membrane, production of exopolysaccharide (EPS), above all in a biofilm state, represent some strategies to counteract AMP activity [183]. However, these mechanisms are nonspecific, and, to date, AMPs have gained much attention in the treatment of biofilm-associated infection. Indeed, biofilm represents an ecological niche where different microbial populations, bearing a variety of ARGs, coexist and promote HGT mechanisms [184]. To this regard, examples of AMPs, such as BLFPs, were able to interfere with cellular mechanisms involved in biofilm formation by the foodborne *P. fluorescens;* indeed, BLFP treatment determined a reduction of biofilm biomass, EPS production, and pigment synthesis under low temperature [21]. Other examples of the efficacy of AMP as antibiofilm agents were reported for *P. aeruginosa* [185,186]. 

Based on these findings, the research for novel antimicrobials has been addressed towards the discovery of compounds acting as inhibitors of QS (QSI), the cell-to-cell communication system regulating the biofilm formation as well as AR [187]. The systematic approaches widely used to identify novel QSI are essentially two. The first is based on the construction of a virulence or resistance reporter system [188] and the identification of small molecules by high-throughput screening. The second is based on the determination of genes and protein responsible for the alteration in the QS mechanism in mutant strains (by screening insertion mutant libraries) and modelling molecules for inhibiting the enzymatic activities of these enzymes, targets for developing new therapeutic strategies [189].

Ding et al. [190,191] applied a molecular docking approach to screen potential QSIs from traditional Chinese medicine and food-derived compound databases (more than 13,000 molecules each), able to interfere with a three-dimensional (3D) structure of LuxI- and LuxR-type regulators; the latter was built by homology modelling using the amino acids sequences obtained through WGS of a *P. fluorescens* strain, causing spoilage of chilled raw milk and aquatic products. Results showed that benzyl alcohol, catechin, (-)-epicatechin, propyl gallate, hesperidin, and lycopene were identified as potent QSIs, and may be applied in food preservation and biofilm elimination. Similarly, in silico molecular docking analysis revealed that cinnamaldehyde, major constituent of Chinese cinnamon, could bind to the *P. fluorescens* LuxR-type protein by hydrogen bonding, suggesting this to be the molecular mechanism used by this molecule to inhibit QS [192].

Although demonstrated as promising strategies, QSI identified in essential oils (such as *trans*-cinnamaldehyde, hexanal, carvacrol, and thymol; [193,194]) have a pungent odor, low solubility, and toxicity that limit their use as food preservatives; thus, recent current research are focused on embedding these compounds into nanoparticles in order to improve their application in food processing or in the development of new packaging technologies [195]; synthesis of natural QSI derivatives with enhanced activity are also ongoing [196].

A very fascinating approach has come from the utilization of bacteriophages, currently applied within therapy for the treatment of *P. aeruginosa* infections involving biofilms on burn patients [197]. The renewed interest in this therapy approach is due to the recent finding of lytic bacteriophage MKO1 (family *Myoviridea*), a naturally occurring virus that forces a desired genetic trade-off between phage resistance and antibiotic sensitivity of *P. aeruginosa* [198]. Indeed, MKO1 utilizes as a receptor-binding site the outer membrane porin M (OprM) of the multidrug efflux systems MexAB and MexXY used by *P. aeruginosa* to extrude antibiotics. In turn, *P. aeruginosa* naturally evolves resistance to phage attack, downregulating the biosynthesis of efflux systems causing it to become increasingly antibiotic sensitive [198]. Although the use of bacteriophages is certified at a clinical level with pathogen-specific commercial drugs [199], unfortunately they are not fully considered in the field of food production, except for the trouble they create in microbial starter cultures. However, bacteriophages could become a potential solution to a variety of issues such as the detection and biocontrol of various undesirable bacteria that cause either infectious diseases or spoilage of food materials [200].

Recently, physical methods, such as High Intensity Light Pulses (HILP), X-ray, and the combination of ultrasound and steam, have been successfully explored for controlling pseudomonads growth and preserving food quality of cold stored mozzarella cheeses [201,202,203]; promising results of these studies highlighted the need to further investigate these innovative technologies in order to be applied in industrial scale-up.

While waiting for these innovative control strategies to be optimized and become economically sustainable for industry, the prevention of microbial contamination by *Pseudomonas* spp. spoilers in dairy environment is essential to contain bacterial mobile resistome. Thus, in addition to conventional strategies [204,205], cost-effective and eco-friendly food-processing technologies are catching on. Among these, a preserving acidified brine, composed by calcium lactate and lactic acid, has been used to delay *Pseudomonas* spp. and *Enterobacteriaceae* growth, prolonging mozzarella cheese shelf life without impairing the sensory characteristics of cheese; experimental brine also prevented a blue discoloration appearance on cheese surface [206]. 

Other treatments include ozone or electrolyzed water, successfully used for the removal of milk residues and biofilm-forming bacteria (such as *P. fluorescens*) from stainless steel surfaces and in milk processing, including fluid milk, powdered milk products, and cheese [207,208]. The latter have been proven as effective alternative strategies for the elimination of *P. fluorescens* biofilm, which are also resistant to conventional sanitizers (e.g., sodium hypochlorite; [209]).

## 6. Concluding Remarks

The boost of MDR bacteria spread is a worldwide threat that needs urgent action especially because it is dramatically impairing antibiotics effectiveness and consequently may bode very high morbidity and mortality levels associated with infections. The struggle against this issue is currently carried out on several fronts mostly by synthesizing new antimicrobial drugs, counteracting inappropriate use or abuse of antibiotics in agriculture, environment, animal, and human medicine, and identifying molecular mechanisms underlying ARGs acquisition and expression in human pathogens or environmental saprophytes of clinical interest. Over time, many studies have focused on MDR foodborne pathogens and commensals, mostly occurring in foods and the food chain. However, environmental saprophytic bacteria and especially those that co-inhabit the boundaries between the environment and the food system represent the main ARGs reservoir. In this context, the spoilage microbiota of refrigerated fresh dairy products (e.g., mozzarella cheese) comprises several psychrotrophic bacterial species, such as *Pseudomonas* spp., that take advantage of the storage conditions and grow reaching high loads. With the evidence of detrimental effects on food shelf-life caused by these bacteria, this review has summarily brought forth the underestimated issue of MDR psychrotrophic spoilage pseudomonads naturally occurring in retailed fresh dairy products. For the first time, here, cheese-associated resistome of psychrotrophic pseudomonads bearing ARGs for β-lactams and carbapenems was shown. To date, WGS analyses have partially defined the taxonomic placement of this highly versatile bacterial group of SSO, but metagenomic studies will enable us to evaluate their actual spread in dairy farms and across the fresh dairy product chain, elucidating the contamination routes and eventual drivers, and identifying mobile genetic determinants associated with ARGs. Concurrently, as highlighted in this review, novel and attractive antimicrobial strategies, mostly addressed to spoilage reduction, could be exploited in addition to the appropriate cleaning procedures in order to counteract the growth of psychrotrophic pseudomonads in fresh dairy products during cold storage, and inhibit biofilm formation. 

## Figures and Tables

**Figure 1 foods-08-00372-f001:**
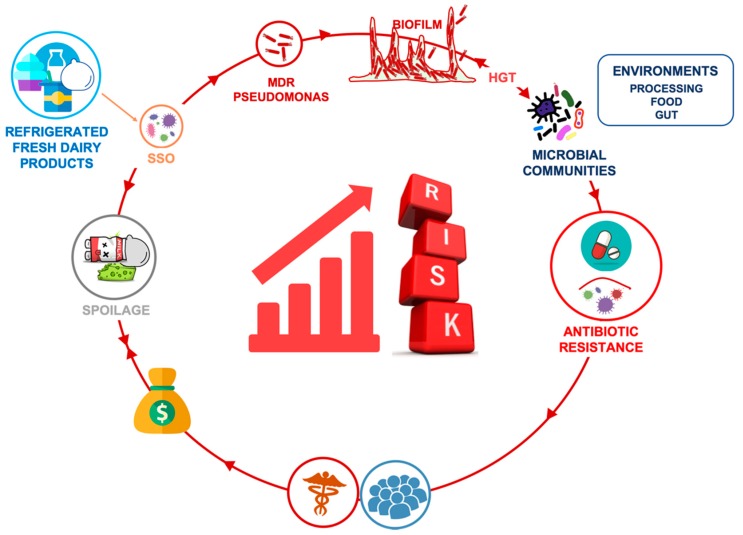
Routes of spread and impacts of antibiotic resistance of psychrotrophic pseudomonads spoiling refrigerated fresh dairy products and spreading throughout different environments.

**Figure 2 foods-08-00372-f002:**
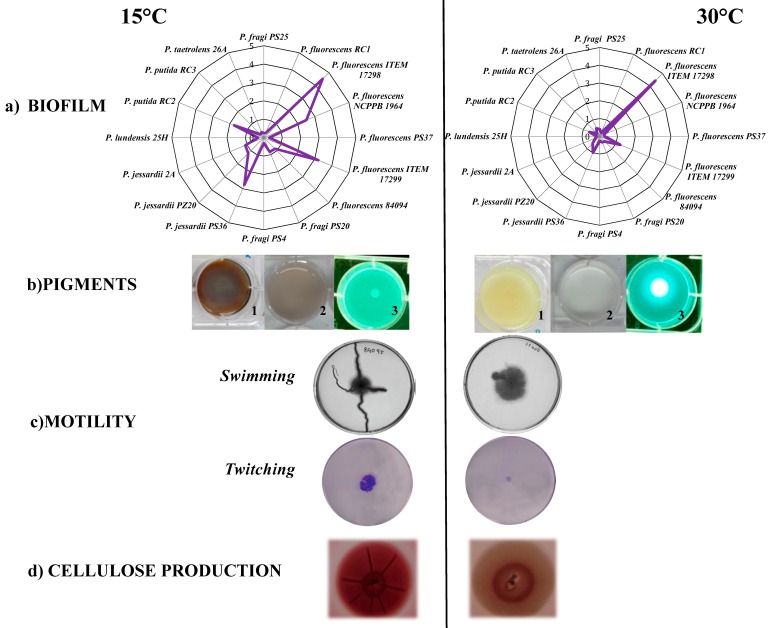
Phenotypic characteristics associated with biofilm formation by *Pseudomonas* spp., isolated from dairy cheese [20,41], under two temperatures of incubation (15 and 30 °C). (**a**) Radar plots of biofilm forming *Pseudomonas* spp. after 72 h of incubation at two temperature of incubation 15 and 30 °C. Values from 0 to 5 represent absorbance of Cristal Violet (CV) at 570 nm measured as reported by Quintieri et al. [21]; (**b**) Pigment production by *P. fluorescens* ITEM 17298 inoculated in Luria Bertani agar (1), M63 broth (2) and Pseudomonas Agar Base (3) after 48 h of incubation. For (3) resulting colonies were examined under Wood’s lamp; (**c**) Swimming and twitching motility of *P. fluorescens* ITEM 17298 inoculated on M8 with 0.3% or 1% of agar, respectively. For twitching biofilm formation on the bottom of plate dish was determined as previously described [21]; (**d**) Cellulose production (red crystals) by *P. fluorescens* ITEM 17298 on Congo Red Agar [147].

**Table 1 foods-08-00372-t001:** List of *Pseudomonas* spp. isolated from dairy foods and resistant to different classes of antibiotics.

Species	Source	Antibiotics		References
		Class	Molecule (μg)	
*P. pseudoalcaligenes*, *P. fluorescens biovar V*, *P. alcaligenes*, *P. pseudoalcaligenes subspecies citrulli*	Turkish homemade white cheese	β-lactams and β-lactams/β-lactamas inhibitors	Penicillin G (10 μg); Piperacillin (16 μg); Piperacillin/tazobactam (64/4 μg);	[37]
		Sulfanilamide/2,4-diaminopyrimidine	Sulfamethoxazole/trimethoprim (25 μg)	
*P. fluorescens*, *P. tolaasii*	Raw milk from Finland farms	β-lactams and β-lactams/β-lactamas inhibitors	Ticarcillin (64 μg); Ticarcillin-Clavulonic acid (64/2 μg);	[84]
		Monocyclic bacterially derived beta-lactam	Aztreonam (32 μg);		
		Phosphonic acid derivative	Fosfomycin (32 μg);	
		Third-generation cephalosporins	Ceftazidim (32 μg);	
		Aminoglycosides	Tobramycin (8 μg); Amikacin (16 μg); Netilmicin (4 μg), Gentamicin (8 μg);	
		Fluoroquinolones	Ofloxacin (1 μg); ciprofloxacin (4 μg);	
		Lipopeptides	Colistin (2 μg);	
		Sulfanilamide/2,4-diaminopyrimidine	Sulfamethoxazole/trimethoprim (2/38 μg)	
*P. putida*, *P. fulva*, *P. fragi*, *P. mosselii*, *P. rhodesiae*, *P. libanensis*, *P. teatrolens*, *P. chlororaphis*, *P. fluorescens*	Italian bulk tank milk	Penicillin	Piperacillin (100 ug), Ticarcillin/ clavulanic acid (85 ug);	[85]
		Monocyclic bacterially derived beta-lactam	Aztreonam (30 μg);	
		Third and Fourth-generation cephalosporin	Ceftazidim (30 μg); Cefepime (30 μg);	
		Aminoglycosides	Tobramycin (10 μg); Amikacin (30 μg); Netilmicin (10 μg);	
		Fluoroquinolones	Ciprofloxacin (5 μg); Levofloxacin (5 μg);	
		Carbapenems	Meropenem (10 μg); Imipenem (10 μg);	
		Lipopeptides	Colistin sulphate (10 μg);	
*P. fluorescens*, *P. taetrolens*, *P. putida*, *P. fragi*, *P. alcaligenes*, *P. lundensis*	French milks or semi-hard and soft, smear-ripened cheeses	Penicillin	Ticarcillin (75 μg); Amoxicillin (25 μg); Ampicillin (10 μg); Mecillinam (10 μg); Amoxicillin/Clavulanic acid (20/10 μg);	[86]
		Monocyclic bacterially derived beta-lactam	Aztreonam (30 μg);	
		First and Third-generation cephalosporin	Cefalotin (30 μg); Cefotaxime (30 μg);	
		Lipopeptides	Colistin sulphate (50 μg);	
		Polyketide antibiotics	Tetracycline (30 μg);	
		Amphenicol-class	Chloramphenicol (30 μg);	
*Pseudomonas spp.*	Commercial UHT milk	Monocyclic bacterially derived beta-lactam	Aztreonam	[87]
		Carbapenems	Meropenem	
		Aminoglycosides	Amikacin; Gentamicin	
		Third and Fourth generation cephalosporins	Ceftazidime; Cefepime	
		Fluoroquinolones	Levofloxacin	
*P. fluorescens*, *P. gessardii*, *P. fragi*	Italian high moisture mozzarella cheese	Aminoglycosides	Tobramycin (10 μg); Kanamycin (30 μg); Gentamicin (10 μg); Streptomycin (10 µg);	This work (see Table 2)
		Fluoroquinolones	Ofloxacin (5 μg); Ciprofloxacin (5 μg);	
		Quinolones	Nalidixic acid (30 µg);	
		Nitrofurans	Nitrofurantoin (300 µg)	

**Table 2 foods-08-00372-t002:** Antibiotic susceptibility of food-spoilage *Pseudomonas* spp. strains by disk diffusion technique on Muller–Hinton agar according to EUCAST guidelines version 6.0. Blank disks were used as negative control. The plates were incubated overnight at 30 °C. The images of the plates were digitized and the calibrated area of inhibition halos around each antibiotic disk was measured using the UTHSCSA Image tool for Windows ver. 3.0. *P. fluorescens* NCCP1964, isolated from clogged tap water filter was used as reference strains of the assay. Data are expressed as median (minimum–maximum; *n* = 3).

	*P. fluorescens*				*P. gessardii*	*P. fragi*
	ITEM 17299	ITEM 17298	NCCP 1964	ITEM 17295	PS36	PS4
Ampicillin (10 μg) **	N.D.*	N.D.	N.D.	N.D.	N.D.	N.D.
Methicillin (10 μg)	N.D.	N.D.	N.D.	N.D.	N.D.	N.D.
Oxacillin (1 μg)	N.D.	N.D.	N.D.	N.D.	N.D.	N.D.
Penicillin G (10 μg)	N.D.	N.D.	126 (117–131)	N.D.	N.D.	N.D.
Ceftizoxime (30 μg)	N.D.	N.D.	N.D.	N.D.	N.D.	N.D.
Gentamicin (10 μg)	286 ^a^ (266–297)	N.D.	632 ^c^ (588–657)	318 ^a^ (296–331)	502 ^b^ (467–522)	545 ^b^ (507–567)
Tobramicin (10 μg)	355 ^a^ (330–369)	352 ^a^ (328–367)	678 ^d^ (631–705)	424 ^b^ (394–441)	544 ^c^ (506–566)	457 ^b^ (425–475)
Kanamicin (30 μg)	80 ^a^ (75–84)	161 ^b^ (179–167)	424 ^d^ (394–441)	502 ^e^ (467–522)	544 ^e^ (507–565)	326 ^c^ (303–339)
Ciprofloxacin (5 μg)	443 ^a^ (412–461)	405 ^a^ (377–422)	776 ^d^ (721–807)	678 ^c^ (631–705)	726 ^c^ (675–755)	458 ^a^ (426–476)
Ofloxacin (5 μg)	314 ^a^ (292–327)	611 ^d^ (568–636)	776 ^e^ (723–810)	632 ^d^(588–657)	544 ^c^ (506–566)	435 ^b^(405–452)
Streptomycin (10 μg)	85 ^b^ (79–88)	76 ^a^ (70–79)	462 ^e^ (430–481)	85 ^b^ (79–88)	424 ^e^ (394–441)	377 ^d^ (351–392)
Nalidixic acid (30 μg)	256 ^d^ (238–266)	186 ^c^ (173–193)	387 ^e^ (260–402)	173 ^c^ (161–180)	148 ^b^ (138–154)	235 (218–244)
Tetracycline (30 μg)	N.D.	N.D.	N.D.	N.D.	N.D.	N.D.
Vancomycin (30 μg)	N.D.	N.D.	N.D.	N.D.	N.D.	N.D.
Clindamycin (2 μg)	N.D.	N.D.	N.D.	N.D.	N.D.	N.D.
Lincomycin (2 μg)	N.D.	N.D.	N.D.	N.D.	N.D.	N.D.
Erythromycin (15 μg)	N.D.	N.D.	N.D.	N.D.	N.D.	N.D.
Fusidic acid (10 μg)	N.D.	N.D.	N.D.	N.D.	N.D.	N.D.
Nitrofurantoin (300 μg)	N.D.	N.D.	35 (33–37)	N.D.	N.D.	N.D.

* N.D.: not detected. ** Different letters for each extract in a row show statistically significant differences (*p* < 0.05) between medians in non-normal distribution (Kruskal–Wallis test; medians were compared by Dunn’s test).

**Table 3 foods-08-00372-t003:** Antibiotic resistance genes identified in *Pseudomonas* spp. (other than *P. aeruginosa*) from different sources.

Species	Source	Antibiotic Resistance Genes	Reference
*P. corrugata*	tomato (Italy)	*arp*C	[96]
*P. monteilii*	clinical isolate	β-*lactamase*, aac(6′)-Ib, sul1	[97]
*P. protegens*	*Pf-5* cotton rhizosphere	*rpo*B mutation	[98]
*P. syringae*	-	*fos*C	[99]
*P. cannabina*	plant	β-*lactamase*, multidrug efflux system transmembrane protein	[100]
*P. chlororaphis*	clinical isolate	β-*lactamase*, ampC, *mbl,* phnP, *cme*ABC, *mex*CD-*opr*J, *mex*E-*opr*N, *fos*E	[101]
*P. citronellolis*	soil collected under pine trees in northern Virginia, USA	β-*lactamase*, *tet*A, o*pr*M1-5, *van*X, *fos*A	[102]
*P. fluorescens*	Industrial strain pre-filtered tanks England	*upp*P, *mex*AB	[103]
	feces of of Mareca penelope	*cme*ABC, *mex*C-m*ex*D-*opr*J, m*ex*E-*mex*F-o*pr*N, *mac*A, *mac*B	[104]
	Siene river	β-*lactamase*	[105]
	clinical isolate		[82]
	coastal water		[106]
	*P. fluorescens* cLP6a (petroleum-contaminated soil)	*emh*ABC	[107]
	wheat take-all decline soil in China		[108]
	*P. fluorescens* cLP6a (petroleum-contaminated soil)		[90]
	mozzarella	*mexA, ampC, oprD, mdtL, emrB*	[95]
	meat microbiome	*tolC*, *mdtB*	[109]
	clinical isolate	*aac*A31, *fos*E, β-*lactamase*	[110]
*P. lutea*	rhizosphera	*amp*G, *amp*E, *amp*D, *mrc*A, *mrc*B, β-*lactamase*, *pbp*C, *mdr*A, *acr*B, *mex*B, *ade*J, *sme*E, *mtr*D, *cme*B, *mar*C, *mdt*C, *mdt*B, *bcr*, *fsr*	[111]
*P. luteola*	clinical isolate	β*-lactamase*	[112]
*P. mosselii*	lower respiratory tract patient	β*-lactamase*, *aacA4*, *aphA15*, and *aadA1*	[113]
*P. otitidis*	food (chicken and pork)	β-*lactamase*	[83]
	clinical isolates		[114]
*P. pseudoalcaligens*	Guadalquivir River	RND efflux pump	[115]
*P. putida*	polluted Nigerian wetlands	β-*lactamase*	[116]
	clinical isolate		[81]
	clinical isolate		[82]
	clinical isolate	β-*lactamase*, *qnr*VC6, *gcu*173, *str*A, *str*B, *aac*A4	[117]
	clinical isolate	β-*lactamase*, *fos*E, *aac*A4, *aad*A1, *dfr*B1b	[118]
	clinical isolate	β-*lactamase*, *aad*A1, *aph*(3′)-XV, *aac*A4, *aph*3-Ib, *str*A, *str*B, *sul*1	[119]
	toluene enrichment	*ttg*ABC	[91,120]
	S12 from soil isolated styrene enrichment	*arp*ABC	[121]
	B6 soil	*ttg*ABC, *srp*ABC, *ttg*GHI	[122]
	B6 soil	30 efflux pump coding genes	[123]
	clinical isolate	*aad*A2, *qac*ED1, *sul*1	[124]
	clinical isolate	β-*lactamase*, *pnr*VC6, *aac*A3, ISP*pu*24, *cat*B11c, *Gcu*56, *aad*A1a, *dfr*B2c, *aac*A4′, *cat*B3	[124]
	clinical isolate	RND pumps, *cme*ABC, MATE family efflux pump	[125]
	clinical isolate	*ttg*GHI, β*-lactamase*, *ttg*ABC, *sul*1, *str*A, *mer*A, *tet*A, *aph*A1-IAB, *aad*A1, *ttg*GHI	[126]
	clinical isolate	β*-lactamase*, *aac*A4	[127]
	shrimp	*qnrA, qnrB*	[70]
*P. stutzeri*	clinical isolate	β-*lactamase*	[128]
	clinical isolate		[129]
	*P. stutzeri* strain ZoBell (ATCC14405) marine sample taken in the Pacific Ocean	MATE efflux pump	[130]
	clinical isolate	β-*lactamase*, *aac*A7, *dfr*B5 gene, *aac*C-A5	[131]
*P. syringae*	*P. syringae pv. syringae* B728a snap bean leaflet in Wisconsin, and *P. syringae pv. tomato* DC3000	RND-type multidrug efflux pump, *mex*AB-*opr*M	[132]
*P. syringae* pv. *actinidiae*	*Actinidia* pathogen	resistance nodulation division (RND), multi antimicrobial resistance (MAR), multidrug endosomal transporter (MET), major facilitator superfamily (MFS)	[133]
*P. syringae* pv. *syringae*	snap bean leaflet in Wisconsin	*str*A, *str*B	[134]
	plant		[135]
*P. xanthomarina*	Strain UASWS0955 sewage sludge	fosmidomycin, polymyxin, penicillin, fluoroquinolones resistance genes (not specified)	[136]

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
