# Peer review of "Antibiotic Resistant Pseudomonas Spp. Spoilers in Fresh Dairy Products: An Underestimated Risk and the Control Strategies"

_foods, 2019, doi:10.3390/foods8090372_

Round 1
Reviewer 1 Report
For better assimilation of antibiotic resistance threat to the public health, the reviewer suggests utilizing information provided by the U.S. CDC’s report on the antimicrobial resistance threat to the public health in the beginning part of the introduction. This 2013 report, an example could be utilized (and/or newer complementary studies of the agency):
https://www.cdc.gov/drugresistance/threat-report-2013/pdf/ar-threats-2013-508.pdf
The reviewer also suggests a discussion on vertical and horizontal gene transfer mechanisms in general and how these mechanisms could lead to the development of inter and intra-species resistance to antimicrobial chemotherapies. This could further provide justification that while MDR in this particular species is not directly a public health threat, but very well could lead to the transfer of the resistant genes via horizontal gene transfer mechanisms to pathogenic agents. This could enhance the assimilation of the study by the Foods readership if incorporated early in the introduction section.
Although this is not a book chapter and extensive discussion of nomenclature and taxonomy of the Pseudomonas spp. is not necessarily needed, but the reviewer suggests a brief section on the organisms nomenclature and natural microbiota of dairy products, in general, to assure assimilation of the manuscript content by the stakeholders.
The manuscript has some level of identical property to published literature, the reviewer suggests using the accompany similarity index report carefully and eliminating sections that are identical to previously published literature.
Author Response
For better assimilation of antibiotic resistance threat to the public health, the reviewer suggests utilizing information provided by the U.S. CDC’s report on the antimicrobial resistance threat to the public health in the beginning part of the introduction. This 2013 report, an example could be utilized (and/or newer complementary studies of the agency):
https://www.cdc.gov/drugresistance/threat-report-2013/pdf/ar-threats-2013-508.pdf
Thank you for your suggestion, we included few sentences in the introduction to cite the report.
The reviewer also suggests a discussion on vertical and horizontal gene transfer mechanisms in general and how these mechanisms could lead to the development of inter and intra-species resistance to antimicrobial chemotherapies. This could further provide justification that while MDR in this particular species is not directly a public health threat, but very well could lead to the transfer of the resistant genes via horizontal gene transfer mechanisms to pathogenic agents. This could enhance the assimilation of the study by the Foods readership if incorporated early in the introduction section.
Thank you, we add several lines in the introduction to generally describe vertical and horizontal gene transfer and the impact of HGT in the spread of ARG in diary.
Although this is not a book chapter and extensive discussion of nomenclature and taxonomy of the Pseudomonas spp. is not necessarily needed, but the reviewer suggests a brief section on the organisms nomenclature and natural microbiota of dairy products, in general, to assure assimilation of the manuscript content by the stakeholders.
We think that the second paragraph exhaustively covers the taxonomy of Pseudomonas genus and its recent update. In addition, table 1 reported the list of Pseudomonas spp. isolated from dairy foods and resistant to different class of antibiotics. We do not want to focus the attention of the reader on taxonomic or nomenclature aspects to the detriment of the major topic of the manuscript, which is the antibiotic resistance and its spread in the diary sector.
The manuscript has some level of identical property to published literature, the reviewer suggests using the accompany similarity index report carefully and eliminating sections that are identical to previously published literature.
We are sure that the topic discussed in this review is really new and original. However, we examined the text by using free available software (such as checktext.org t and https://plagiarismdetector.net/Plagiarism Checker by Grammarly) to detect any plagiarism but we did not found evidence of this. Nevertheless, this is a review, so it is not unusual to cite published literature, and provide the source for the citation, as we always did.
Reviewer 2 Report
Esteemed Authors,
It has been a great honor, as well as a pleasantly challenging activity, to review the article entitled ”Antibiotic resistant Pseudomonas spp. spoilers in fresh dairy products: an underestimated risk and the control strategies”.
Within the above-mentioned article, the authors have chosen to provide a detailed picture of the importance of antibiotic-resistant Pseudomonas spp. for the food chain, especially for the fresh dairy products.
The article approaches a topic that is highly significant to consumers as well as to public health systems and their evolution, in order to ensure improvement, especially when it comes to the interaction between these systems and public policies in the considered area.
The article is structured following the classic model for this type of article - review, comprising 5 parts: Introduction; Pseudomonas spp. genus and species occurrence in dairy products; Antibiotic resistance in Pseudomonas spp. spoiler: mechanisms and influencing factors in dairy sector; Sequencing-based tools and database for AR prediction; Strategies to control the spread of antibiotic resistant Pseudomonas spp. in dairy sector, and Concluding remarks.
The 5 major components of the article are balanced dimension-wise and are presented coherently and logically, tightly linked to one another. additionally, all five main parts also benefit from the direct scientific support of the authors of the article (I refer to previous research in the field, iconography or other support items).
The article is very well documented, and most bibliographic references are recent and very recent.
I would advise the authors to be more careful with regard to the bibliography: it is preferred that the cited authors be mentioned in alphabetical order, and references without specified authors be mentioned at the end of the list of references, in chronological order. I also recommend using a single system not only in citations but also when it comes to the journals. I am referring here mainly to mentioning the following elements for each article consulted: journal, volume, issue and pages (the DOI may also be mentioned, should the author so desire, but the basic descriptive elements are the previously mentioned ones).
For example: Akbas M.Y. (2015). Bacterial biofilms and their new control strategies in food industry. In: The Battle Against Microbial Pathogens: Basic Science, Technological Advances and Educational Programs (Méndez-Vilas A., Ed.), FORMATEX, pp. 383-394.
Another example: Berendonk T.U., Manaia Célia, Merlin C., Fatta-Kassinos D., Cytryn E., Walsh Fiona, Bürgmann H., Sørum H., Norström Madelaine, Pons Marie-Noëlle, Kreuzinger N., Huovinen P., Stefani Stefania, Schwartz T., Kisand V., Baquero F., Martinez J.L. (2015). Tackling antibiotic resistance: the environmental framework. Nature Reviews Microbiology (or JCR Abbreviation – Nat. Rev. Microbiol.), 13, 310-317.
From this point of view, there are several unsatisfactory situations, as follows:
1. Articles that are repeated - for example – Zhang et al., 2018 – page 30, lines 54-59, numbers 202 and 203 in the bibliographic references list; Ghosh et al., 2019 – page 23, lines 785-786, number 64 in the bibliographic references list, and page 24, lines 829-830, number 82 in the bibliographic references list;
2. Authors that appear in the text but do not appear in the bibliographic references list – for example – Gupta et al., 2014 – page 15, line 381;
3. Authors that appear in the bibliographic references list but do not appear in the text: - for example – Saha B.J., 2018 – page 28, lines 1052-1054, number 162 in the bibliographic references list;
4. Some small inadvertencies between years – for example: Yamamoto et al., 2000, number 198 in the bibliographic references list appear like Yamamoto et al., 2018 in table 3; Winsor et al., 2016, appear like Winsor et al., 2017 (page 15, line 400); Fanelli et al., 2017, number 68 in the bibliographic references list appear like Fanelli et al., 2018 in table 3, and other similar situations;
5. Some small inadvertencies between authors – for example – page 24, lines 873-875, number 97 in the bibliographic references list: Kenzaka T., Tani K. (2018). Draft Genome Sequence of Carbapenem-Resistant Pseudomonas fluorescens Strain BWKM6, Isolated from Feces of Mareca penelope. Genome Announcements, 6(12)e00186-18; DOI: 10.1128/genomeA.00186-18.
6. Authors who do not follow the alphabetical order and are quoted in a place other than the one that is due – for example – Coton et al., 2012 – page 29, lines 1140-1142, number 197 in the references list.
As for the paper grammar, the article is generally very well written: only a few shortcomings in the grammar of the text can be mentioned, as follows:
Page 1, line 36 – replace ‘’of patient’’ with ‘’of a patient’’;
Page 1, line 39 – replace ‘’In last decades’’ with ‘’In the last decades’’;
Page 1, line 39 – replace ‘’has elevated’’ with ‘’elevated’’;
Page 3, line 118 – replace ‘’comprises’’ with ‘’comprise’’;
Page 5, line 189 – replace ‘’quite homogeneous’’ with ‘’homogeneous’’;
Page 5, line 190 – replace ‘’contained’’ with ‘’itcontained’’;
Page 5, line 217 – replace ‘’of problem’’ with ‘’of the problem’’;
Page 13, line 296 – replace ‘’takes place’’ with ‘’occurs’’;
Page 13, line 298 – replace ‘’includes’’ with ‘’include’’;
Page 14, line 345 – replace ‘’has become’’ with ‘’have become’’;
Page 15, line 400 – replace ‘’have contributed’’ with ‘’has contributed’’;
Page 15, line 401 – replace ‘’also integrated’’ with ‘’has also integrated’’;
Page 17, line 476 – replace ‘’Despite conventional’’ with ‘’Despite the conventional’’;
Page 19, line 566 – replace ‘’on going’’ with ‘’ongoing’’;
Page 19, line 610 – replace ‘’these highly’’ with ‘’this highly’’.
As a general conclusion regarding the grammar, the text does not contains other mistakes that need to be corrected. As for editing (writing), the text should be checked once again carefully as there are some small omissions that need to be corrected.
On the whole, the article which is characterized by a high degree of originality, can be considered interesting for academic staff, for researchers in the field and even for the wide public.
Together with other positive elements, the scientific relevance and quality of the presentation will surely make the article attractive to a wide audience, and especially to authors interested in the fields of food microbiology, food safety, food technology, and public health.
Best Regards,
Reviewer
Author Response
It has been a great honor, as well as a pleasantly challenging activity, to review the article entitled ”Antibiotic resistant Pseudomonas spp. spoilers in fresh dairy products: an underestimated risk and the control strategies”.
Within the above-mentioned article, the authors have chosen to provide a detailed picture of the importance of antibiotic-resistant Pseudomonas spp. for the food chain, especially for the fresh dairy products.
The article approaches a topic that is highly significant to consumers as well as to public health systems and their evolution, in order to ensure improvement, especially when it comes to the interaction between these systems and public policies in the considered area.
The article is structured following the classic model for this type of article - review, comprising 5 parts: Introduction; Pseudomonas spp. genus and species occurrence in dairy products; Antibiotic resistance in Pseudomonas spp. spoiler: mechanisms and influencing factors in dairy sector; Sequencing-based tools and database for AR prediction; Strategies to control the spread of antibiotic resistant Pseudomonas spp. in dairy sector, and Concluding remarks.
The 5 major components of the article are balanced dimension-wise and are presented coherently and logically, tightly linked to one another. additionally, all five main parts also benefit from the direct scientific support of the authors of the article (I refer to previous research in the field, iconography or other support items).
The article is very well documented, and most bibliographic references are recent and very recent.
I would advise the authors to be more careful with regard to the bibliography: it is preferred that the cited authors be mentioned in alphabetical order, and references without specified authors be mentioned at the end of the list of references, in chronological order. I also recommend using a single system not only in citations but also when it comes to the journals. I am referring here mainly to mentioning the following elements for each article consulted: journal, volume, issue and pages (the DOI may also be mentioned, should the author so desire, but the basic descriptive elements are the previously mentioned ones).
For example: Akbas M.Y. (2015). Bacterial biofilms and their new control strategies in food industry. In: The Battle Against Microbial Pathogens: Basic Science, Technological Advances and Educational Programs (Méndez-Vilas A., Ed.), FORMATEX, pp. 383-394.
Another example: Berendonk T.U., Manaia Célia, Merlin C., Fatta-Kassinos D., Cytryn E., Walsh Fiona, Bürgmann H., Sørum H., Norström Madelaine, Pons Marie-Noëlle, Kreuzinger N., Huovinen P., Stefani Stefania, Schwartz T., Kisand V., Baquero F., Martinez J.L. (2015). Tackling antibiotic resistance: the environmental framework. Nature Reviews Microbiology (or JCR Abbreviation – Nat. Rev. Microbiol.), 13, 310-317.
From this point of view, there are several unsatisfactory situations, as follows:
Articles that are repeated - for example – Zhang et al., 2018 – page 30, lines 54-59, numbers 202 and 203 in the bibliographic references list; Ghosh et al., 2019 – page 23, lines 785-786, number 64 in the bibliographic references list, and page 24, lines 829-830, number 82 in the bibliographic references list; Authors that appear in the text but do not appear in the bibliographic references list – for example – Gupta et al., 2014 – page 15, line 381; Authors that appear in the bibliographic references list but do not appear in the text: - for example – Saha B.J., 2018 – page 28, lines 1052-1054, number 162 in the bibliographic references list; Some small inadvertencies between years – for example: Yamamoto et al., 2000, number 198 in the bibliographic references list appear like Yamamoto et al., 2018 in table 3; Winsor et al., 2016, appear like Winsor et al., 2017 (page 15, line 400); Fanelli et al., 2017, number 68 in the bibliographic references list appear like Fanelli et al., 2018 in table 3, and other similar situations; Some small inadvertencies between authors – for example – page 24, lines 873-875, number 97 in the bibliographic references list: Kenzaka T., Tani K. (2018). Draft Genome Sequence of Carbapenem-Resistant Pseudomonas fluorescens Strain BWKM6, Isolated from Feces of Mareca penelope. Genome Announcements, 6(12)e00186-18; DOI: 10.1128/genomeA.00186-18. Authors who do not follow the alphabetical order and are quoted in a place other than the one that is due – for example – Coton et al., 2012 – page 29, lines 1140-1142, number 197 in the references list.
Thank you, indeed there were some inadvertencies we have carefully revised and correct.
As for the paper grammar, the article is generally very well written: only a few shortcomings in the grammar of the text can be mentioned, as follows:
Page 1, line 36 – replace ‘’of patient’’ with ‘’of a patient’’; Thank you, done.
Page 1, line 39 – replace ‘’In last decades’’ with ‘’In the last decades’’; Thank you, done.
Page 1, line 39 – replace ‘’has elevated’’ with ‘’elevated’’; Thank you, done.
Page 3, line 118 – replace ‘’comprises’’ with ‘’comprise’’; Thank you, done.
Page 5, line 189 – replace ‘’quite homogeneous’’ with ‘’homogeneous’’; Thank you, done.
Page 5, line 190 – replace ‘’contained’’ with ‘’itcontained’’; Thank you, done.
Page 5, line 217 – replace ‘’of problem’’ with ‘’of the problem’’; Thank you, done.
Page 13, line 296 – replace ‘’takes place’’ with ‘’occurs’’; Thank you, done.
Page 13, line 298 – replace ‘’includes’’ with ‘’include’’; Thank you, done.
Page 14, line 345 – replace ‘’has become’’ with ‘’have become’’; Thank you, done.
Page 15, line 400 – replace ‘’have contributed’’ with ‘’has contributed’’; Thank you, done.
Page 15, line 401 – replace ‘’also integrated’’ with ‘’has also integrated’’; Thank you, done.
Page 17, line 476 – replace ‘’Despite conventional’’ with ‘’Despite the conventional’’; Thank you, done.
Page 19, line 566 – replace ‘’on going’’ with ‘’ongoing’’; Thank you, done.
Page 19, line 610 – replace ‘’these highly’’ with ‘’this highly’’. Thank you, done.
As a general conclusion regarding the grammar, the text does not contains other mistakes that need to be corrected. As for editing (writing), the text should be checked once again carefully as there are some small omissions that need to be corrected.
On the whole, the article which is characterized by a high degree of originality, can be considered interesting for academic staff, for researchers in the field and even for the wide public.
Together with other positive elements, the scientific relevance and quality of the presentation will surely make the article attractive to a wide audience, and especially to authors interested in the fields of food microbiology, food safety, food technology, and public health.
Thank you for all your comments and suggestion. We modified the manuscript accordingly.